# EvoGrad: Efficient Gradient-Based Meta-Learning and Hyperparameter Optimization

**Ondrej Bohdal[1], Yongxin Yang[1], Timothy Hospedales[1,2]**
[1] School of Informatics, The University of Edinburgh
[2] Samsung AI Research Centre, Cambridge
{ondrej.bohdal, yongxin.yang, t.hospedales}@ed.ac.uk

## Abstract

Gradient-based meta-learning and hyperparameter optimization have seen significant progress recently, enabling practical end-to-end training of neural networks together with many hyperparameters. Nevertheless, existing approaches are relatively expensive as they need to compute second-order derivatives and store a longer computational graph. This cost prevents scaling them to larger network architectures. We present EvoGrad, a new approach to meta-learning that draws upon evolutionary techniques to more efficiently compute hypergradients. EvoGrad estimates hypergradient with respect to hyperparameters without calculating second-order gradients, or storing a longer computational graph, leading to significant improvements in efficiency. We evaluate EvoGrad on three substantial recent meta-learning applications, namely cross-domain few-shot learning with feature-wise transformations, noisy label learning with Meta-Weight-Net and low-resource cross-lingual learning with meta representation transformation. The results show that EvoGrad significantly improves efficiency and enables scaling meta-learning to bigger architectures such as from ResNet10 to ResNet34.

## 1  Introduction

Gradient-based meta-learning and hyperparameter optimization have been of long-standing interest in neural networks and machine learning [11, 21, 4]. Hyperparameters (aka meta-parameters) can take diverse forms, especially under the guise of meta-learning, where there has recently been an explosion of successful applications addressing diverse learning challenges [9]. For example to name just a few: training optimizer initial condition in support of few-shot learning [7, 1, 15]; training instance-wise weights for cleaning noisy datasets [31, 26]; training loss functions in support of generalisation [14] and learning speed; and training stochastic regularizers in support of cross-domain robustness [33].

Most of these applications share the property that meta-parameters impact validation loss only indirectly through their effect on model parameters, and so computing validation loss gradients with respect to meta-parameters usually leads to the need to compute second-order derivatives, and store longer computational graphs for backpropagation. This eventually becomes a bottleneck to execution time, and – more severely – to scaling the size of the underlying models, given the practical limitation of GPU memory. There has been steady progress in the development of diverse practical algorithms for computing validation loss with respect to meta-parameters [20, 19, 21]. Nevertheless they mostly share some form of the aforementioned limitations. In particular, the majority of recent successful practical applications [31, 33, 14, 2, 5, 17, 30] essentially use some variant of the $T_1 - T_2$ algorithm [20] to estimate the gradient $\frac{\partial \ell_V}{\partial \lambda}$ of validation loss w.r.t. hyperparameters. This approach computes the gradient online at each step of updating the base model $\theta$, and estimates it as $\frac{\partial \ell_V}{\partial \lambda} \approx \frac{\partial \ell_V}{\partial \theta} \frac{\partial^2 \ell_T}{\partial \theta \partial \lambda}$, for training loss $\ell_T$. As with many alternative estimators, this requires second-order derivatives, and extending the computational graph. Besides the additional computation cost, this limits the size of the

35th Conference on Neural Information Processing Systems (NeurIPS 2021).

base model that can be used in a given GPU, since the memory cost of meta-learning is now multiple times the size of vanilla backpropagation. This in turn prevents the application of meta-learning to problems where large state-of-the-art model architectures are required.

To address this issue, we draw inspiration from evolutionary optimization methods [27] to develop EvoGrad, a meta-gradient algorithm that requires no higher-order derivatives and as such is significantly faster and lighter than the standard approaches. In particular, we take the novel view of estimating meta-gradients via a putative inner-loop *evolutionary* update to the base model. As this requires no gradients itself, the meta-gradient can then be computed using first-order gradients alone, and without extending the computational graph – leading to efficient hyperparameter updates. Meanwhile for efficient and accurate base model learning, the real inner-loop update can separately be carried out by conventional gradient descent.

Our EvoGrad is a general meta-optimizer applicable to many meta-learning applications, among which we choose three to demonstrate its impact: the LFT model [33] observes that a properly tuned stochastic regularizer can significantly improve cross-domain few-shot learning performance. We show that by training those regularizer parameters with EvoGrad, rather than the standard second-order approach, we can obtain the same improvement in accuracy with significant reduction in time and memory cost. This allows us to scale LFT from the original ResNet10 to ResNet34 within a 12GB GPU. Second, the Meta-Weight-Net (MWN) [31] model deals with label noise by meta-learning an auxiliary network that re-weights instance-wise losses to down-weight noisy instances and improve validation loss. We also show that EvoGrad can replicate MWN results with significant cost savings. Third, we demonstrate the benefits of EvoGrad on an application from NLP, in addition to the ones from computer vision: low-resource cross-lingual learning using MetaXL approach [35].

To summarize, our main contributions are: (1) We introduce EvoGrad, a novel method for gradient-based meta-learning and hyperparameter optimization that is simple to implement and efficient in time and memory requirements. (2) We evaluate EvoGrad on a variety of illustrative and substantial meta-learning problems, where we demonstrate significant compute and memory benefits compared to standard second-order approaches. (3) In particular, we illustrate that EvoGrad allows us to scale meta-learning to bigger models than was previously possible on a given GPU size, thus bringing meta-learning closer to the state-of-the-art frontier of real applications. We provide source code for EvoGrad at `https://github.com/ondrejbohdal/evograd`.

## 2   Related work

Gradient-based meta-learning solves a bilevel optimization problem where validation loss is optimized with respect to the meta-knowledge by backpropagating through the update of the model on training data and with meta-knowledge. The meta-knowledge updates form an outer loop, around an inner loop of base model updates. The inner loop can run for one [20], few [29, 21], or many [19] steps within each outer-loop iteration. Meta-knowledge can take many forms, for example, it can be an initialization of the model weights [7], feature-wise transformation layers [33], regularization to improve domain generalization [2] or even a synthetic training set [34, 5]. Most substantial practical applications use a one or few-step inner loop for efficiency.

More recently, several methods [19, 25] have utilized Implicit Function Theorem (IFT) to develop new gradient-based meta-learners. These methods use multiple inner-loop steps without the need to backpropagate through whole inner loop, which significantly improves memory efficiency over methods that need keep track of the whole inner loop training process. However, IFT methods assume the model has converged in the inner loop. This makes them unsuited for the majority of practical applications above where training the inner loop to convergence for each hypergradient step is infeasible. Furthermore, the hypergradient is still more costly compared to one-step $T_1 - T_2$ method. The costs come from the associated overhead with approximating an inverse Hessian of the training data with respect to the model parameters. Note that the Hessian itself does not need to be stored due to the mechanics of reverse-mode differentiation [8, 3]. However, this does not eliminate the remaining calculations which still require higher-order gradients that result in backpropagation via longer graphs due to additional gradient nodes. For these reasons, we focus comparison on the more widely used $T_1 - T_2$ strategy which is oriented at single-step inner loops similar to EvoGrad.

Theoretically it is also possible to use hypernetworks [18] to find good hyperparameters in a first-order way. Hypernetworks take hyperparameters as inputs and generate model parameters. However, the

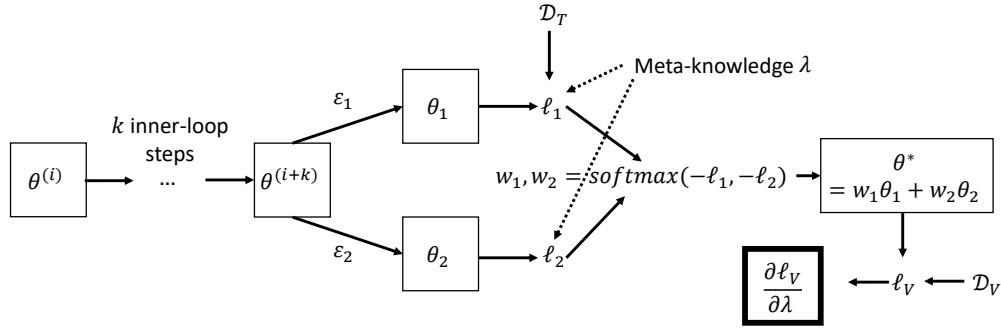

Figure 1: Graphical illustration of a single EvoGrad update using $K = 2$ model copies.

approach is not commonly used, likely due to the difficulty of generating well-performing model parameters. We provide experimental results to support this hypothesis in the supplementary material.

Meta-learning can be categorized into several groups, depending on the type of meta-knowledge and also if the model is trained from scratch as part of the inner loop [9]. Offline meta-learning approaches train a model from scratch per each update of the meta-knowledge, while online meta-learning approaches train the model and meta-knowledge jointly. As a result, offline meta-learning is extremely expensive [6, 37] when scaled beyond few-shot learning problems where only a few iterations are sufficient for training [7, 1, 15]. Therefore most larger-scale problems [16, 10, 33, 35] use online learning in practice, and this is where we focus our contribution.

The meta-knowledge to learn can take different forms. A particular dichotomy is between the special case where the meta-knowledge corresponds to the base model itself, in the form of an initialization; and the more general cases where it does not. The former initialization meta-learning has been popularized by MAML [7], and is widely used in few-shot learning. This can be solved relatively efficiently, for example using a first-order approximation of MAML [7], Reptile [23] or minibatch proximal update [36]. On the other hand, there are vastly more cases [9] where the meta-knowledge is different from the model itself, such as LFT's stochastic regularizer to improve cross-domain generalization [33], MWN's instance-wise loss weighting network for label noise robustness [31], a label generation network to improve self-supervised generalization [17], a Feature-Critic loss to improve domain generalization [14] and many others. In this more general case, most applications rely on a $T_1 - T_2$-like algorithm, as the efficient approximations specific to MAML do not apply. The ability to significantly improve the efficiency of gradient-based meta-learning would have a large impact as methods like these would directly benefit from it in runtime and energy consumption. More crucially, they could scale to bigger and more state-of-the-art neural network architectures.

## 3 Methods

### 3.1 Background: meta-learning as bilevel optimization

We aim to solve a bilevel optimization problem where our goal is to find hyperparameters $\boldsymbol{\lambda}$ that minimize the validation loss $\ell_V$ of the model parametrized by $\boldsymbol{\theta}$ and trained with loss $\ell_T$ and $\boldsymbol{\lambda}$:

$$\boldsymbol{\lambda}^* = \arg\min_{\boldsymbol{\lambda}} \ell_V^*(\boldsymbol{\lambda}), \text{ where } \ell_V^*(\boldsymbol{\lambda}) = \ell_V(\boldsymbol{\lambda}, \boldsymbol{\theta}^*(\boldsymbol{\lambda})) \text{ and } \boldsymbol{\theta}^*(\boldsymbol{\lambda}) = \arg\min_{\boldsymbol{\theta}} \ell_T(\boldsymbol{\lambda}, \boldsymbol{\theta}). \quad (1)$$

In order to meta-learn the value of $\boldsymbol{\lambda}$ using gradient-based methods, we need to calculate the hypergradient $\frac{\partial \ell_V}{\partial \boldsymbol{\lambda}}$. We can expand its calculation as follows:

$$\frac{\partial \ell_V^*(\boldsymbol{\lambda})}{\partial \boldsymbol{\lambda}} = \frac{\partial \ell_V(\boldsymbol{\lambda}, \boldsymbol{\theta}^*(\boldsymbol{\lambda}))}{\partial \boldsymbol{\lambda}} + \frac{\partial \ell_V(\boldsymbol{\lambda}, \boldsymbol{\theta}^*(\boldsymbol{\lambda}))}{\partial \boldsymbol{\theta}^*(\boldsymbol{\lambda})} \frac{\partial \boldsymbol{\theta}^*(\boldsymbol{\lambda})}{\partial \boldsymbol{\lambda}}. \quad (2)$$

In meta-learning and hyperparameter optimization more broadly, the direct term $\frac{\partial \ell_V(\boldsymbol{\lambda}, \boldsymbol{\theta}^*(\boldsymbol{\lambda}))}{\partial \boldsymbol{\lambda}}$ is typically zero because the hyperparameter does not directly influence the value of the validation loss – it influences it via the impact on the model weights $\boldsymbol{\theta}$. However, the model weights $\boldsymbol{\theta}$ are themselves trained using gradient optimization, which gives rise to higher-order derivatives. We propose a variation on this step where the update of the model weights is inspired by evolutionary methods, allowing us to eliminate the need for higher-order derivatives. We consider the setting where the hypergradient of hyperparameter $\lambda$ is estimated online [20] together with updating the base model $\theta$, as this is the most widely used setting in substantial practical applications [31, 33, 14, 2, 5, 30, 16, 35].

### 3.2 The EvoGrad update

Given the current model parameters $\theta \in \mathbb{R}^M$, hyperparameters $\lambda \in \mathbb{R}^N$, training loss $\ell_T$ and validation loss $\ell_V$, we aim to estimate $\frac{\partial \ell_V}{\partial \boldsymbol{\lambda}}$ for efficient gradient-based hyperparameter learning. The key idea is – solely for the purpose of hypergradient estimation – to consider a simple evolutionary rather than gradient-based inner-loop step on $\theta$.

**Evolutionary inner step**    First, we sample random perturbations $\boldsymbol{\epsilon} \in \mathbb{R}^M \sim \mathcal{N}(\mathbf{0}, \sigma \boldsymbol{I})$, and apply them to $\boldsymbol{\theta}$. Sampling $K$ perturbations, we can create a population of $K$ variants $\{\boldsymbol{\theta}_k\}_{k=1}^K$ of the current model as $\boldsymbol{\theta}_k = \boldsymbol{\theta} + \boldsymbol{\epsilon}_k$. We can now compute the training losses $\{\ell_k\}_{k=1}^K$ for each of the $K$ models, $\ell_k = f(\mathcal{D}_T | \boldsymbol{\theta}_k, \boldsymbol{\lambda})$ using the current minibatch $\mathcal{D}_T$ drawn from the training set. Given these loss values, we can calculate the weights (sometimes called fitness) of the population of candidate models as

$$w_1, w_2, \ldots, w_K = \mathrm{softmax}([-\ell_1, -\ell_2, \ldots, -\ell_K]/\tau), \tag{3}$$

where $\tau$ is a temperature parameter that rescales the losses to control the scale of weight variability.

Given the weights $\{w_k\}_{k=1}^K$, we complete the current step of evolutionary learning by updating the model parameters via the affine combination

$$\boldsymbol{\theta}^* = w_1 \boldsymbol{\theta}_1 + w_2 \boldsymbol{\theta}_2 + \cdots + w_K \boldsymbol{\theta}_K. \tag{4}$$

**Computing the hypergradient**    We now evaluate the updated model $\boldsymbol{\theta}^*$ for a minibatch from the validation set $\mathcal{D}_V$ and take gradient of the validation loss $\ell_V = f(\mathcal{D}_V | \boldsymbol{\theta}^*)$ w.r.t. the hyperparameter:

$$\frac{\partial \ell_V}{\partial \boldsymbol{\lambda}} = \frac{\partial f(\mathcal{D}_V | \boldsymbol{\theta}^*)}{\partial \boldsymbol{\lambda}} \tag{5}$$

One can easily verify that the computation in Eq. 5 does not involve second-order gradients as no first-order gradients were used in the inner loop. This is in contrast to the typical approach [20, 21] of applying gradient-based updates in the inner loop and differentiating through it (in either forward-mode or reverse-mode), or even applying the implicit function theorem (IFT) [19], all of which trigger higher-order gradients and an extended computation graph.

**Algorithm flow**    In practice we follow the flow of $T_1 - T_2$ [20] used by many substantive applications [33, 31, 2, 16, 35]. We take alternating steps on $\theta$ using the exact gradient $\frac{\partial \ell_T}{\partial \boldsymbol{\theta}}$, and on $\boldsymbol{\lambda}$ using the hypergradient $\frac{\partial \ell_V}{\partial \boldsymbol{\lambda}}$, which in EvoGrad is estimated as in Eq. 5.

### 3.3 EvoGrad hypergradient as a random projection

To understand EvoGrad, observe that the hyper-gradient in Eq. 5 expands as

$$\frac{\partial \ell_V}{\partial \boldsymbol{\lambda}} = \frac{\partial \ell_V}{\partial \boldsymbol{\theta}^*} \frac{\partial \boldsymbol{\theta}^*}{\partial \boldsymbol{\lambda}} = \frac{\partial \ell_V}{\partial \boldsymbol{\theta}^*} \mathcal{E} \frac{\partial \boldsymbol{w}}{\partial \boldsymbol{\lambda}} = \frac{\partial \ell_V}{\partial \boldsymbol{\theta}^*} \mathcal{E} \frac{\partial \boldsymbol{w}}{\partial \boldsymbol{\ell}} \frac{\partial \boldsymbol{\ell}}{\partial \boldsymbol{\lambda}} \tag{6}$$

where $\mathcal{E} = [\boldsymbol{\epsilon}_1, \boldsymbol{\epsilon}_2, \ldots, \boldsymbol{\epsilon}_K]$ is the $M \times K$ matrix formed by stacking $\boldsymbol{\epsilon}_k$'s as columns, $\boldsymbol{w}$ is the $K$-dimensional ($\boldsymbol{w} = [w_1, w_2, \ldots, w_K]$) vector of candidate model weights, and $\boldsymbol{\ell} = [\ell_1, \ell_2, \ldots, \ell_K]$ is the $K$-dimensional vector of candidate model losses.

Recall that $\mathcal{E}$ is a random matrix, so the operation $\frac{\partial \ell_V}{\partial \boldsymbol{\theta}^*} \mathcal{E}$ can be understood as randomly projecting the $M$-dimensional validation loss' gradient to a new low-dimensional space of dimension $K \ll M$. Alternatively, we can interpret the update as factorising the model-parameter-to-hyperparameter derivative $\frac{\partial \boldsymbol{\theta}^*}{\partial \boldsymbol{\lambda}}$ (sized $M \times N$) into two much smaller matrices $\mathcal{E}$ and $\frac{\partial \boldsymbol{w}}{\partial \boldsymbol{\lambda}}$ of size $M \times K$ and $K \times N$.

In terms of implementation, $\frac{\partial \ell_V}{\partial \boldsymbol{\theta}^*}$ is obtained by backpropagation and $\mathcal{E}$ is sampled on the fly. The term $\frac{\partial w}{\partial \boldsymbol{\lambda}} = \frac{\partial w}{\partial \ell} \frac{\partial \ell}{\partial \boldsymbol{\lambda}}$ is computed by the softmax-to-logit derivative ($K \times K$) and the derivative of the $K$ candidate models training losses w.r.t. hyperparameters. It is noteworthy that the $K$ elements of $\frac{\partial \ell}{\partial \boldsymbol{\lambda}}$ are completely independent, and can be computed in parallel where multiple GPUs are available.

## 3.4 Comparison to other methods

We compare EvoGrad to the most related and widely-used alternative $T_1 - T_2$ [20] in Table 1. $T_1 - T_2$ requires higher-order gradients and associated longer computational graphs – due to the need to backpropagate through gradient nodes. This leads to increased memory and time cost compared to vanilla backpropagation. In contrast, EvoGrad requires no higher-order gradients, no large matrices, and no substantial expansion of the computational graph.

Table 1: Comparison of hypergradient approximations of $T_1 - T_2$ and EvoGrad.

| Method | Hypergradient approximation |
|---|---|
| $T_1 - T_2$ [20] | $\frac{\partial \ell_V}{\partial \boldsymbol{\lambda}} - \frac{\partial \ell_V}{\partial \boldsymbol{\theta}} \times \boldsymbol{I} \frac{\partial^2 \ell_T}{\partial \boldsymbol{\theta} \partial \boldsymbol{\lambda}^T}$ |
| EvoGrad (ours) | $\frac{\partial \ell_V}{\partial \boldsymbol{\lambda}} + \frac{\partial \ell_V}{\partial \boldsymbol{\theta}} \times \mathcal{E} \frac{\partial w}{\partial \ell} \frac{\partial \ell}{\partial \boldsymbol{\lambda}} = \frac{\partial \ell_V}{\partial \boldsymbol{\lambda}} + \frac{\partial \ell_V}{\partial \boldsymbol{\theta}} \times \mathcal{E} \frac{\partial \operatorname{softmax}(-\boldsymbol{\ell})}{\partial \boldsymbol{\lambda}}$ |

We analyse the asymptotic big-$\mathcal{O}$ time and memory requirements of EvoGrad vs $T_1 - T_2$ in Table 2. The dominant cost in terms of both memory and time is the cost of backpropagation. Backpropagation is significantly more expensive than forward propagation because forward propagation does not need to store all intermediate variables in memory [25, 8]. Note that even if EvoGrad keeps multiple copies of the model weights in memory, this cost is small compared to the cost of backpropagation, and the latter is done with only *one* set of weights $\boldsymbol{\theta}^*$. We remark that our main empirical results are obtained with only $K = 2$ models, so we can safely ignore this in our asymptotic analysis.

In addition, we elaborate on how higher-order gradients contribute to increased memory and time costs. Results from computing the first-order gradients are added into the computation graph as new nodes in the graph so that we can calculate the higher-order gradients. When calculating the higher-order gradients, we backpropagate through this longer computational graph, which directly increases the memory and time costs. The current techniques [20] rely on longer computational graphs, while EvoGrad significantly shortens the graph and reduces memory cost by avoiding this step. This consideration is not visible in the big-$\mathcal{O}$ analysis, but contributes to improved efficiency.

Table 2: Comparison of asymptotic memory and operation requirements of EvoGrad and $T_1 - T_2$ meta-learning strategies. $P$ is the number of model parameters, $H$ is the number of hyperparameters. $K \ll H$ is the number of model copies in EvoGrad. Note this is a first-principles analysis, so the time requirements are different when using e.g. reverse-mode backpropagation that uses parallelization.

| Method | Time requirements | Memory requirements |
|---|---|---|
| $T_1 - T_2$ [20] | $\mathcal{O}(PH)$ | $\mathcal{O}(P + H)$ |
| EvoGrad (ours) | $\mathcal{O}(KP + H)$ | $\mathcal{O}(P + H)$ |

## 4 Experiments

We first consider two simple problems: 1) a 1-dimensional problem where we try to find the minimum of a function, and 2) meta-learning a feature-transformer to find the rotation that correctly aligns images whose training and validation sets differ in rotation. This serves as a proof-of-concept problem to show our method is capable of meta-learning suitable hyperparameters. We then consider three real problems where meta-learning has been used to solve different learning challenges. We show that EvoGrad makes a significant impact in terms of reducing the memory and time costs (while keeping the accuracy improvements brought by meta-learning): 3) Cross-domain few-shot classification via learned feature-wise transformation [33], 4) Meta-Weight-Net: learning an explicit mapping for sample weighting [31], 5) MetaXL: meta representation transformation for low-resource cross-lingual learning [35]. We provide a brief overview of each problem, together with evaluation and analysis. Further details and experimental settings are described in the supplementary material.

## 4.1 Illustration using a 1-dimensional problem

In this problem we minimize a validation loss function $f_V(x) = (x - 0.5)^2$ where parameter $x$ is optimized using SGD with training loss function $f_T(x) = (x - 1)^2 + \lambda\|x\|_2^2$ that includes a meta-parameter $\lambda$. A closed-form solution for the hypergradient is available and is equal to $g(\lambda) = (\lambda - 1)/(\lambda + 1)^3$, which allows us to compare EvoGrad against the ground-truth gradient.

Our first analysis studies the estimated EvoGrad hypergradient for a grid of $\lambda$ values between 0 and 2. For each value of $\lambda$ we show the mean and standard deviation of the estimated $\partial f_V/\partial\lambda$ over 100 repetitions (with random choice of $x$). We use temperature $\tau = 0.5$, $\epsilon \in \mathbb{R} \sim \mathcal{N}(0,1)$ and consider between 2 and 100 models copies in the population. The results in Figure 2 show that EvoGrad estimates have a similar trend to the ground-truth gradient, even if the EvoGrad estimates are noisy. The level of noise decreases with more models in the population, but the correct trend is visible even if we use only 2 models.

Our second analysis studies the trajectories that parameters $x, \lambda$ follow if they are both optimized online using SGD with learning rate of 0.1 for 5 steps, starting from five different positions (circles). The hypergradients are either estimated using EvoGrad or directly using the ground-truth formula. Figure 3 shows that the trajectories of both variations are similar, and they become more similar as we use more models in the population. In all cases the parameters converge towards the lightly-coloured region where the validation loss is the lowest at $x = 0.5$.

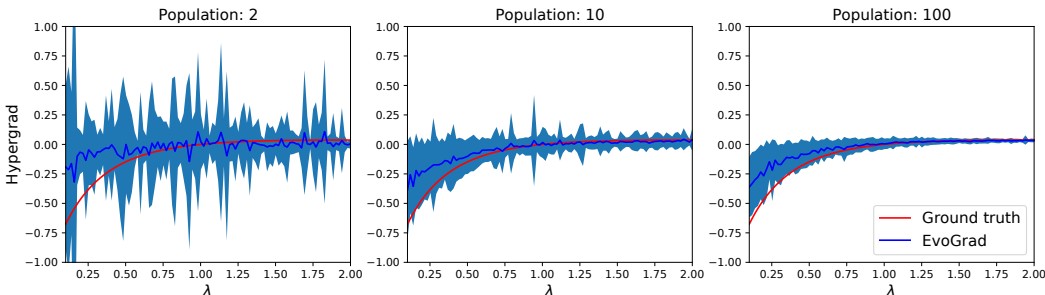

Figure 2: Comparison of the hypergradient $\partial f_V/\partial\lambda$ estimated by EvoGrad vs the ground-truth.

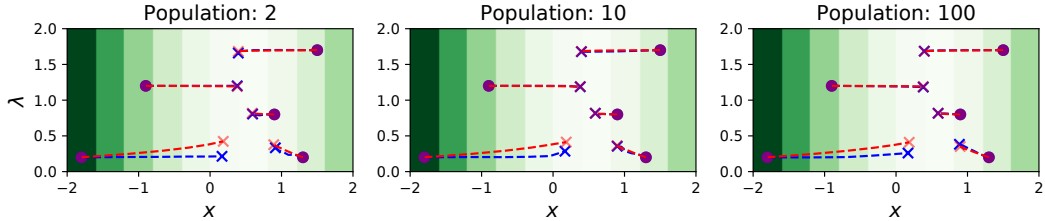

Figure 3: Trajectories of parameters $x, \lambda$ when following $\partial f_T/\partial x$ and $\partial f_V/\partial\lambda$ using SGD for 5 random starting positions. Comparison of trajectories using EvoGrad estimated (blue) or ground-truth (red) hypergradient. The initial position is marked with a circle, and the final position after 5 steps is marked with a cross. The shading is validation loss $f_V(x)$.

## 4.2 Rotation transformation

In this task we work with MNIST images [12], and assume that the validation and test sets are rotated by $30°$ compared to the conventionally oriented training images. Clearly, directly training a model and applying it will lead to low performance. We therefore assume meta-knowledge in the form of a hidden rotation. The rotation transformation is applied to the training images before learning, and should itself be meta-learned by the validation loss obtained by the CNN trained on the rotated training set. Thus solving the meta-learning problem should result in a $30°$ rotation, and a base CNN that generalises to the rotated validation set.

The problem is framed as online meta-learning where each update of the base model is followed by a meta-parameter update using EvoGrad. We use EvoGrad with 2 model copies, temperature $\tau = 0.05$ and $\sigma = 0.001$ for $\epsilon \sim \sigma\text{sign}(\mathcal{N}(\mathbf{0}, \mathbf{I}))$. Our LeNet [13] base model is trained for 5 epochs.

We repeat the experiments 5 times and show a comparison of the results in Table 3. A baseline model achieves $98.40 \pm 0.07\%$ accuracy if the test images are not rotated, but its accuracy drops to $81.79 \pm 0.64\%$ if the same images are rotated by $30°$. A model trained with EvoGrad and the rotation transformer is able to accurately classify rotated images, with a similar accuracy as the baseline model can classify unrotated images. This confirms we can successfully optimize hyperparameters with EvoGrad. The meta-learned rotation is also close to the true value.

Table 3: Rotation transformation learning. The goal is to accurately classify MNIST test images rotated by $30°$ degrees compared to the training set orientation. Test accuracies (%) of a baseline model, and one whose training set has been rotated by the EvoGrad's meta-learned rotation, and associated EvoGrad rotation estimate ($°$). Accuracy for rotation matched train/test sets is $98.40\%$.

| True Rotation | Baseline Acc. | EvoGrad Acc. | EvoGrad Rotation Est. |
|---|---|---|---|
| $30°$ | $81.79 \pm 0.64$ | $98.11 \pm 0.32$ | $28.47° \pm 5.23°$ |

### 4.3 Cross-domain few-shot classification via learned feature-wise transformation

As the next task we consider cross-domain few-shot classification (CD-FSL). CD-FSL is considered an important and highly challenging problem at the forefront of computer vision. The state-of-the-art approach learned feature-wise transformation (LFT) [33] aims to meta-learn stochastic feature-wise transformation layers that regularize metric-based few-shot learners to improve their few-shot learning generalisation in cross-domain conditions. The method includes two key steps: 1) updating the model with the meta-parameters on a pseudo-seen task and 2) updating the meta-parameters by evaluating the model on a pseudo-unseen task by backpropagating through the first step. As feature-wise transformation is not directly used for the pseudo-unseen task, this leads to higher-order gradients. Note that the problem itself is memory-intensive because we work with larger images of size $224 \times 224$ within episodic learning tasks. As a result, a significantly more efficient meta-learning approach could allow us to scale from the ResNet10 model used in the paper to a larger model.

We experiment with the LFT-RelationNet [32] metric-based few-shot learner and consider the exact same experiment settings as [33] using the official PyTorch implementation associated with the paper. LFT introduces 3712 hyper-parameters to train for ResNet10, and 9344 for ResNet34. All our experiments are conducted on Titan X GPUs with 12GB of memory using $K = 2$ for EvoGrad.

Table 4 shows the baseline performance of vanilla unregularised ResNet (-), manually tuned FT layers (FT), FT layers meta-learned by second-order gradient (LFT) and by EvoGrad. The results show that EvoGrad matches the accuracy of the original LFT approach, leading to clear accuracy improvements over training with no feature-wise transformation or training with fixed feature-wise parameters selected manually. At the same time EvoGrad is significantly more efficient in terms of the memory and time costs as shown in Figure 4. The memory improvements from EvoGrad allow us to scale the base feature extractor to ResNet34 within the standard 12GB GPU. The original LFT with its $T_1 - T_2$ style second-order algorithm cannot be extended in the available memory if we keep the same settings of the few-shot learning tasks. Thus, we are able to improve state-of-the-art accuracy on both 5-way 1 and 5-shot tasks. For ResNet34, we include baselines without any feature-wise transformation and with manually chosen feature-wise transformation to confirm the benefit of meta-learning.

### 4.4 Label noise with Meta-Weight-Net

We consider a further highly practical real problem where online meta-learning has led to significant improvements – learning from noisy labelled data. The Meta-Weight-Net framework trains an auxiliary neural network that performs instance-wise loss re-weighting on the training set [31]. The base model is updated using the sum of weighted instance-wise losses for noisy data, while the Meta-Weight-Net itself is updated by evaluating the updated model on clean validation data and by backpropagating through the model update. We use the official implementation of the approach [31] and follow the same experimental settings, using $K = 2$ for EvoGrad.

Table 4: RelationNet test accuracies (%) and 95% confidence intervals across test tasks on various unseen datasets. LFT EvoGrad can scale to ResNet34 on all tasks within 12GB GPU memory, while vanilla second-order LFT $T_1 - T_2$ cannot. We also report the results of our own rerun of the LFT approach using the official code – denoted as *our run*. EvoGrad can clearly match the accuracies obtained by the original approach that uses $T_1 - T_2$.

|  | Model | Approach | CUB | Cars | Places | Plantae |
|---|---|---|---|---|---|---|
| 5-way 1-shot | ResNet10 | - | $44.33 \pm 0.59$ | $29.53 \pm 0.45$ | $47.76 \pm 0.63$ | $33.76 \pm 0.52$ |
|  |  | FT | $44.67 \pm 0.58$ | $30.38 \pm 0.47$ | $48.40 \pm 0.64$ | $35.40 \pm 0.53$ |
|  |  | LFT $T_1 - T_2$ | $48.38 \pm 0.63$ | $32.21 \pm 0.51$ | $50.74 \pm 0.66$ | $35.00 \pm 0.52$ |
|  |  | LFT $T_1 - T_2$ (our run) | $46.03 \pm 0.60$ | $31.50 \pm 0.49$ | $49.29 \pm 0.65$ | $36.34 \pm 0.59$ |
|  |  | LFT EvoGrad | $47.39 \pm 0.61$ | $32.51 \pm 0.56$ | $50.70 \pm 0.66$ | $36.00 \pm 0.56$ |
|  | ResNet34 | - | $45.61 \pm 0.59$ | $29.54 \pm 0.46$ | $48.87 \pm 0.65$ | $35.03 \pm 0.54$ |
|  |  | FT | $45.15 \pm 0.59$ | $30.28 \pm 0.44$ | $49.96 \pm 0.66$ | $35.69 \pm 0.54$ |
|  |  | LFT EvoGrad | $45.97 \pm 0.60$ | $33.21 \pm 0.54$ | $50.76 \pm 0.67$ | $38.23 \pm 0.58$ |
| 5-way 5-shot | ResNet10 | - | $62.13 \pm 0.74$ | $40.64 \pm 0.54$ | $64.34 \pm 0.57$ | $46.29 \pm 0.56$ |
|  |  | FT | $63.64 \pm 0.77$ | $42.24 \pm 0.57$ | $65.42 \pm 0.58$ | $47.81 \pm 0.51$ |
|  |  | LFT $T_1 - T_2$ | $64.99 \pm 0.54$ | $43.44 \pm 0.59$ | $67.35 \pm 0.54$ | $50.39 \pm 0.52$ |
|  |  | LFT $T_1 - T_2$ (our run) | $65.94 \pm 0.56$ | $43.88 \pm 0.56$ | $65.57 \pm 0.57$ | $51.43 \pm 0.55$ |
|  |  | LFT EvoGrad | $64.63 \pm 0.56$ | $42.64 \pm 0.58$ | $66.54 \pm 0.57$ | $52.92 \pm 0.57$ |
|  | ResNet34 | - | $63.33 \pm 0.59$ | $40.50 \pm 0.55$ | $64.94 \pm 0.56$ | $50.20 \pm 0.55$ |
|  |  | FT | $62.48 \pm 0.56$ | $41.06 \pm 0.52$ | $64.39 \pm 0.57$ | $50.08 \pm 0.55$ |
|  |  | LFT EvoGrad | $66.40 \pm 0.56$ | $44.25 \pm 0.55$ | $67.23 \pm 0.56$ | $52.47 \pm 0.56$ |

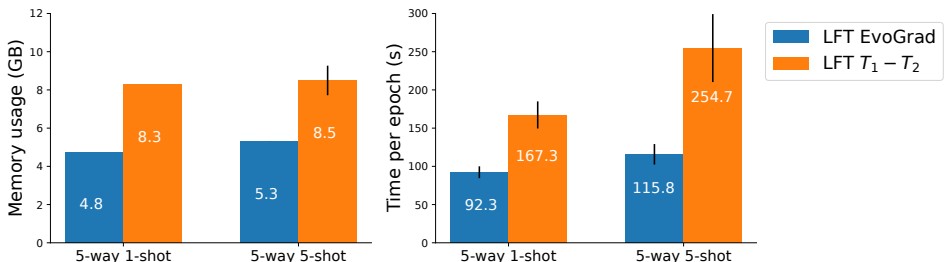

Figure 4: Cross-domain few-shot learning with LFT [33]: analysis of memory and time efficiency of EvoGrad vs standard second-order $T_1 - T_2$ approach. Mean and standard deviation reported across experiments with different test datasets. EvoGrad is significantly more efficient in terms of both memory usage and time per epoch.

Our results in Table 5 confirm we replicate the benefits of training with Meta-Weight-Net, clearly surpassing the accuracy of the baseline when there is label noise. We also note that EvoGrad can improve the accuracy over the $T_1 - T_2$-based approach because the two approaches are distinct and provide different estimates of the true hypergradient. Figure 5 shows that our method leads to significant improvements in memory and time costs (over half of the memory is saved and the runtime is improved by about a third).

Table 5: Test accuracies (%) for Meta-Weight-Net label noise experiments with ResNet-32 – means and standard deviations across 5 repetitions for the original second-order algorithm vs EvoGrad. EvoGrad is able to match or even exceed the accuracies obtained by the original MWN approach.

| Dataset | Noise rate | Baseline | MWN $T_1 - T_2$ | MWN $T_1 - T_2$ (our run) | MWN EvoGrad |
|---|---|---|---|---|---|
| CIFAR-10 | 0% | $92.89 \pm 0.32$ | $92.04 \pm 0.15$ | $91.10 \pm 0.19$ | $92.02 \pm 0.31$ |
|  | 20% | $76.83 \pm 2.30$ | $90.33 \pm 0.61$ | $89.31 \pm 0.40$ | $89.86 \pm 0.64$ |
|  | 40% | $70.77 \pm 2.31$ | $87.54 \pm 0.23$ | $85.90 \pm 0.45$ | $87.74 \pm 0.54$ |
| CIFAR-100 | 0% | $70.50 \pm 0.12$ | $70.11 \pm 0.33$ | $68.42 \pm 0.36$ | $69.16 \pm 0.49$ |
|  | 20% | $50.86 \pm 0.27$ | $64.22 \pm 0.28$ | $63.43 \pm 0.43$ | $64.05 \pm 0.63$ |
|  | 40% | $43.01 \pm 1.16$ | $58.64 \pm 0.47$ | $56.54 \pm 0.90$ | $57.44 \pm 1.25$ |

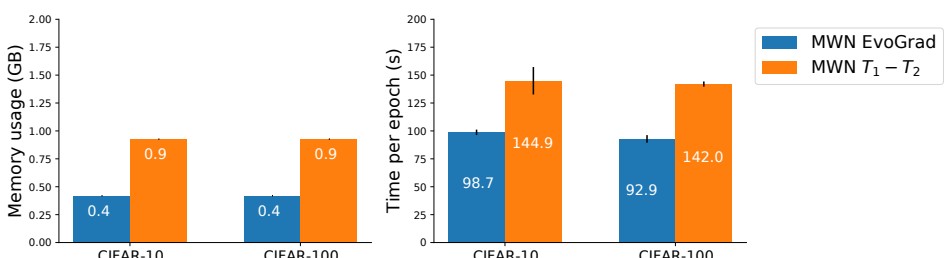

Figure 5: Analysis of memory and time cost of MWN EvoGrad vs the original second-order MWN, showing significant efficiency improvements by EvoGrad. Mean and standard deviation is reported across 5 repetitions of 40% label noise problem.

## 4.5 Low-resource cross-lingual learning with MetaXL

The previous two real applications of meta-learning considered computer vision problems. To highlight EvoGrad is a general method that can make an impact in any domain, we also demonstrate its benefits on a meta-learning application from NLP. More specifically, we use EvoGrad for MetaXL [35], which meta-learns meta representation transformation to better transfer from source languages to low-resource target languages.

We have selected the named entity recognition (NER) task with English source language (WikiAnn dataset [24]), which is one of the key experiments in the MetaXL paper [35]. Table 6 shows EvoGrad matches and in fact surpasses the average test F1 score of MetaXL with the original $T_1 - T_2$ meta-learning method. Figure 6 shows EvoGrad significantly improves both memory and time consumption compared to MetaXL $T_1 - T_2$. Overall these results confirm EvoGrad is suitable for meta-learning in various domains, including both computer vision and NLP.

Table 6: Test F1 score in % for named entity recognition task. English source language. The first two rows are taken from the MetaXL paper, while our own runs are in the following rows. EvoGrad clearly matches and even surpasses the performance of $T_1 - T_2$ baseline. Joint-training (JT) represents a simple non-meta-learning baseline approach.

| Method | qu | cdo | ilo | xmf | mhr | mi | tk | gn | Average |
|---|---|---|---|---|---|---|---|---|---|
| JT | 66.10 | 55.83 | 80.77 | 69.32 | 71.11 | 82.29 | 61.61 | 65.44 | 69.06 |
| MetaXL $T_1 - T_2$ | 68.67 | 55.97 | 77.57 | 73.73 | 68.16 | 88.56 | 66.99 | 69.37 | 71.13 |
| JT (our run) | 59.75 | 49.19 | 79.43 | 68.85 | 68.42 | 89.94 | 61.90 | 69.44 | 68.37 |
| MetaXL $T_1 - T_2$ (our run) | 65.29 | 56.33 | 76.50 | 67.24 | 71.17 | 89.41 | 66.67 | 64.11 | 69.59 |
| MetaXL EvoGrad | 71.00 | 57.02 | 85.99 | 70.40 | 65.45 | 88.12 | 66.97 | 70.91 | 71.98 |

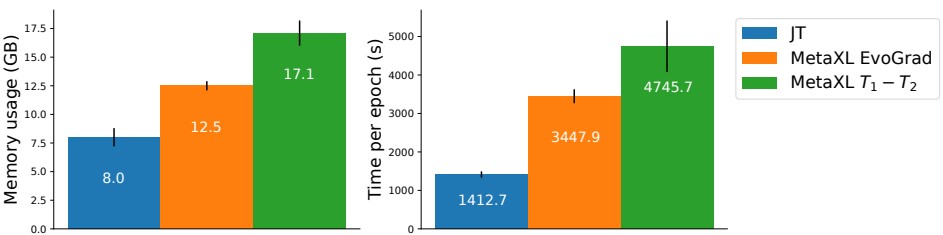

Figure 6: Analysis of memory and time cost of MetaXL EvoGrad vs the original second-order MetaXL, in the context of a simple joint-training (JT) baseline. EvoGrad consumes significantly less memory than $T_1 - T_2$ and is faster. Mean and standard deviation is calculated over the 8 different target languages.

### 4.6 Scalability analysis

We use the Meta-Weight-Net benchmark to study how the number of model parameters affects the memory usage and training time of EvoGrad, comparing it to the standard second-order $T_1 - T_2$ approach. We vary model size by changing the number of filters in the original ResNet32 model, multiplying the filter number $\times 1, \ldots, \times 5$. The smallest model had around 0.5M parameters and the largest one around 11M parameters.

The results in Figure 7 show our EvoGrad leads to significantly lower training time and memory usage, and that the margin over the standard second-order optimizer grows as the model becomes larger. Further, we have analysed the impact of modifying the number of hyperparameters – from 300 up to 30,000. The impact on memory and time was negligible, and both remained roughly constant, which is caused by the main model being significantly larger. It is also because of the fact that reverse-mode differentiation costs scale with the number of model parameters rather than hyperparameters [22] – recall that backpropagation is the main driver of memory and time costs [25]. Moreover, we have done experiments that varied the number of model copies in EvoGrad. The results showed the training time per epoch increased slightly, while the memory costs remained similar.

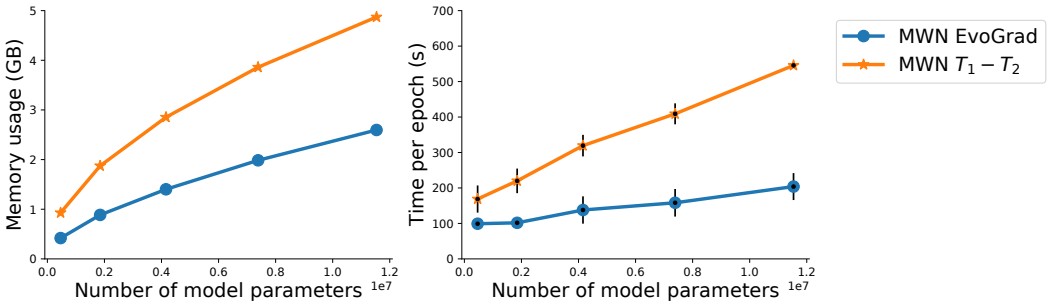

Figure 7: Memory and time scaling of MWN EvoGrad vs original second-order Meta-Weight-Net. Efficiency margins of EvoGrad are larger for larger models.

## 5 Discussion and limitations

Similar to many other gradient-based meta-learning methods, our method is greedy as it considers only the current state of the model when updating the hyperparameters – rather than the whole training process. However, this greediness allows the method to be used in larger-scale settings where we train the hyperparameters and the model jointly. Further, our method approximates the hypergradient stochastically. While results were good for the suite of problems considered here using only $K = 2$, the gradient estimates may be too noisy in other applications. This could lead to poor outcomes which could be a problem in socially important applications. Alternatively, it may necessitate using a larger model population (Figure 2). While as we observed in Section 3.3 the candidate models can be trivially parallelized to scale population size, this still imposes a larger energy cost [28]. Another limitation is that similarly to IFT-based estimators [19], EvoGrad is not suitable for optimizing learner hyperparameters such as learning rate. Currently we have used the simplest possible evolutionary update in the inner loop, and upgrading EvoGrad to a state-of-the-art evolutionary strategy may lead to better gradient estimates and improve results further.

## 6 Conclusions

We have proposed a new efficient method for meta-learning that allows us to scale gradient-based meta-learning to bigger models and problems. We have evaluated the method on a variety of problems, most notably meta-learning feature-wise transformation layers, training with noisy labels using Meta-Weight-Net, and meta-learning meta representation transformation for low-resource cross-lingual learning. In all cases we have shown significant time and memory efficiency improvements, while achieving similar or better performance compared to the existing meta-learning methods.

## Acknowledgments and Disclosure of Funding

This work was supported in part by the EPSRC Centre for Doctoral Training in Data Science, funded by the UK Engineering and Physical Sciences Research Council (grant EP/L016427/1) and the University of Edinburgh.

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
