# OpenReview forum: "EvoGrad: Efficient Gradient-Based Meta-Learning and Hyperparameter Optimization"
_NeurIPS.cc/2021/Conference — NeurIPS 2021 Poster_

### Official Review · Reviewer_ULem · 2021-06-30

**Rating:** 6
**Confidence:** 5

**Summary:**

The paper proposes to replace gradient descent with evolutionary search in the inner loop of meta-learning. This makes the computation of the outer loop first-order optimization, which saves compute and memory compared to second-order optimization.

**Limitations And Societal Impact:**

Please see the main review.

**Main Review:**

The writing is very clear, and the problem the paper is solving is also well-motivated: the cost of second-order optimization required in meta-learning makes it prohibitive for many real-world applications.

The idea is original, but the experimental results in the paper do not convince the reader about its significance. The authors mention that the so-called T1-T2 approach is often used as a first-order approximation to calculating the second-order meta gradients, and remark that EvoGrad is asymptotically more efficient. While it might be more efficient to take a single EvoGrad step compared to a gradient step, we might need a lot more EvoGrad steps compared to gradient steps (in fact, there are also classical theoretical results to this effect [1]). This means that in practice, it is likely that EvoGrad is not as efficient as T1 - T2.

The authors should show experimental comparisons with T1 - T2 to convince the reader that EvoGrad is more efficient. I would also recommend benchmarking on standard meta-learning datasets like OmniGlot or MiniImageNet to increase the significance of the results.

Edit: In my original review, I did not understand that the authors were targeting their work specifically for settings where first-order gradient approximations do not exist, e.g. hyperparameter optimization. This is quite different from experimental benchmarks on OmniGlot/MiniImageNet where MAML-like algorithms are commonly being evaluated on. At my request, the authors added in experiments that show that their evolutionary algorithm based approach significantly outperforms a gradient based approach like a linear hypernet. I'm changing my score to a 6 to reflect this.

[1]: Query Complexity of Derivative-Free Optimization. Kevin G. Jamieson, Robert Nowak, Ben Recht.

**Time Spent Reviewing:**

3

---

> ### Author Response · Authors · 2021-08-10
> **Response to the preliminary review**
>
> We appreciate your effort spent reviewing and providing feedback on our paper, and your comments on its motivation and novelty. We would like to point out a key misunderstanding in the review: it is not true that we do not compare to T1-T2 in the paper. In fact, we compare T1-T2 throughout. Furthermore, we already control the number of iterations used by EvoGrad and T1-T2 and, contrary to the reviewer’s conjecture, EvoGrad is actually significantly faster in terms of the time it takes to do the total number of iterations.
>
> **T1-T2 comparison:** T1-T2 is the underlying meta-learning algorithm that is used by the original MetaWeightNet [29] and Cross-domain few-shot classification via learned feature-wise transformation [31] (this directly follows from our discussion in the Related work section). Our main experiments are exactly to replace the existing T1-T2 hypergradient estimator in these benchmarks with our EvoGrad estimator. Our thorough and in-depth comparison shows the clear benefits of EvoGrad. Of course, if we did not compare with T1-T2, it would be a clear reason for rejection. Since we rigorously compare T1-T2 and surpass it in efficiency (Fig 4, 5, 6, etc.), we believe the reviewer’s main concern is addressed, and the score should be significantly increased. We will make it more explicit in the writing and figure legends to avoid confusion to the reader.
>
> **Iterations to convergence:** The reviewer is concerned that EvoGrad may require more steps than a standard meta-learning approach such as T1-T2. In fact our experiments already control for the number of iterations. We use exactly the same number of meta-gradient update steps for EvoGrad and T1-T2 as we simply replace the optimizers within the original benchmarks [29, 31]: thus EvoGrad does converge as quickly as T1-T2 in terms of iterations. And our paper’s results already make it clear that EvoGrad is more efficient in terms of memory and compute-per-epoch. Thus the wall-clock-time to converge for EvoGrad reflects the improved wall-clock-time per epoch (Fig 5). EG: Total time of around 5500 vs 8500 seconds for EvoGrad vs T1-T2 respectively, on MWN/CIFAR-100. In the final version we will include learning curves that make this clear.
>
> Finally we note the negative theoretical result [1] mentioned by the reviewer does not apply to our case as it is restricted to derivative-free methods under a noisy function evaluation setting (they do not dispute that derivative-free optimization can be efficient under noise-free evaluation). Our problem is not noisy in this sense, which would correspond to inexact loss evaluation. However, in common with most deep learning, we always get exact loss evaluations. Differently to this, the “noise” in EvoGrad is used to perturb parameter space (as in evolutionary methods), and is thus related to stochastic exploration of directions and step size for finite differences. Furthermore, we note that our hypergradient estimator (see Tab 1/Eq 6) mostly makes use of exact derivatives, and only a single carefully chosen evolutionary approximation to one term.
>
> **Omniglot and MiniImageNet:** The mentioned benchmarks Omniglot and MiniImageNet focus on few-shot learning (FSL), which is a special case of more general hyperparameter meta-learning, where one commonly meta-optimizes the initial condition of a gradient-based learner [7]. EvoGrad is a general hyperparameter optimizer that can also apply to this case. The reason that we do not focus on this application in the paper is that FSL is the special case of meta-learning that already admits efficient first-order solutions (FO-MAML and Reptile). Therefore, as discussed on L92-105, we are interested in evaluating EvoGrad in more general meta-learning applications that do not already have efficient solutions. (We do have results for EvoGrad on these benchmarks, and it performs with similar accuracy & efficiency as the first-order approaches. We can include these results in the final appendix.)
>
> In summary, we believe the rating in this review should be significantly increased because we have clearly addressed all concerns, which were based on misunderstandings.

---

> > ### Comment · Reviewer_ULem · 2021-08-16
> > **Apologies for the misunderstanding!**
> >
> > I wish to apologize to the authors for a significant misunderstanding on my part. I was thinking about this piece of work relative to MAML, where it's clear that a first-order gradient approximation exists. That was the T1-T2 that I was thinking about, rather than Luketina et al. But I realize now that the authors intend to target this piece of work for more general contexts like hyperparameter optimization where this gradient might not exist.
> >
> > That said, I remain unconvinced about the significance of the experimental results.
> >
> > Can't the random projection hypergradient in equation 6. be done with a linear hypernetwork as in [1]? This avoids the cost of second-order optimization, and at the same time, does not have the drawbacks of slower derivative-free optimization. If you are able to do an empirical comparison with a linear hypernetwork like in [1] in terms of wall-clock training time and performance (the memory use should be similar since both approaches are linear projections, you can also use a similar "factorized" approach), that will be extremely convincing for me.
> >
> > I'm not sure I understand your argument about why your proposed approach does not qualify as a noisy function evaluation setting, since the use of mini-batches in deep learning pretty much guarantees that it's a noisy function evaluation.
> >
> >
> > [1] Stochastic Hyperparameter Optimization through Hypernetworks. Lorraine et al.

---

> > > ### Author Response · Authors · 2021-08-23
> > > **Hypernetwork experiments**
> > >
> > > **Linear hypernetwork:** Thanks for the suggestion. We agree that hypernetworks [A] in principle provide an alternative that also avoids second-order gradients. However, this does not reduce the significance of EvoGrad because: (1) Optimization-based approaches like T1-T2 are by far the most widely used approach for meta-learning in practice. The majority of methods in the recent survey [9] are optimization-based, using T1-T2 in particular; while few are based on hypernetworks. Therefore an efficient approach to optimization-based hyperparameter tuning is of immediately high impact. (2) While HyperNet works OK for simpler toy problems such as those in [A], in many sophisticated applications of meta-learning, it may not be possible to express the network parameters as a simple function of the hyperparameters, which would manifest in many applications being difficult to optimize and suffering from poor local minima compared to optimization-based approaches. Issue (2) may explain the limited use of hypernetworks in practice (1).
> > >
> > > We conducted experiments applying the hypernetwork approach (with factorization) to cross-domain few-shot classification via learned feature-wise transformation (LFT). We also verified our hypernet implementation using similar experiments as used in [A] - it successfully replicates the simple experimental results in [A]. The results show that the
> > > HyperNet and EvoGrad approach are both comparably efficient compared to T1-T2. However the HyperNet fails to discover a good solution within the standard number of iterations used throughout, and performance is poor. Tables with the results follow.
> > >
> > > 5-way 5-shot with RelationNet and ResNet10 backbone: memory consumption (GB), time per epoch (s)
> > > and test accuracies (%) with 95% confidence intervals across test tasks on various unseen datasets
> > >
> > > | Approach | Memory |Time per epoch| CUB              | Cars             |  Places          |  Plantae         |
> > > | :------- |:------:|:------------:|:----------------:|:----------------:|:----------------:|:----------------:|
> > > | HyperNet | 4.7GB  |  88.4s       | 56.91 $\pm$ 0.57 | 40.64 $\pm$ 0.56 | 56.08 $\pm$ 0.58 | 44.73 $\pm$ 0.57 |
> > > | T1-T2    | 8.5GB  | 254.7s       | 65.94 $\pm$ 0.56 | 43.88 $\pm$ 0.56 | 65.57 $\pm$ 0.57 | 51.43 $\pm$ 0.55 |
> > > | EvoGrad  | 5.3GB  | 115.8s       | 64.63 $\pm$ 0.56 | 42.64 $\pm$ 0.58 | 66.54 $\pm$ 0.57 | 52.92 $\pm$ 0.57 |
> > >
> > >
> > >
> > > 5-way 1-shot with RelationNet and ResNet10 backbone: memory consumption (GB), time per epoch (s)
> > > and test accuracies (%) with 95% confidence intervals across test tasks on various unseen datasets
> > >
> > > | Approach | Memory |Time per epoch| CUB              | Cars             |  Places          |  Plantae         |
> > > | :------- |:------:|:------------:|:----------------:|:----------------:|:----------------:|:----------------:|
> > > | HyperNet | 4.4GB  |  77.0s       | 38.94 $\pm$ 0.57 | 30.10 $\pm$ 0.48 | 38.07 $\pm$ 0.58 | 33.83 $\pm$ 0.58 |
> > > | T1-T2    | 8.3GB  | 167.3s       | 46.03 $\pm$ 0.60 | 31.50 $\pm$ 0.49 | 49.29 $\pm$ 0.65 | 36.34 $\pm$ 0.59 |
> > > | EvoGrad  | 4.8GB  |  92.3s       | 47.39 $\pm$ 0.61 | 32.51 $\pm$ 0.56 | 50.70 $\pm$ 0.66 | 36.00 $\pm$ 0.56 |
> > >
> > > **Noisy function evaluation setting:** We agree that mini-batching in the context of neural networks can be indeed interpreted as noisy function evaluation. However, the empirical results show that in practice EvoGrad does not suffer from requiring many more optimization steps (as explained in our initial response), so the negative theoretical result does not represent a problem in practice for EvoGrad. Similarly, methods such as Reptile are completely gradient-free, yet work reasonably well in practice as a competitor for MAML. Moreover, this line of discussion does not affect our main point that EvoGrad enables meta-learning to scale to larger models than is possible with gradient-based inner loop.
> > >
> > > [A] Stochastic Hyperparameter Optimization through Hypernetworks. Lorraine et al.

---

> > > > ### Comment · Reviewer_ULem · 2021-08-24
> > > > **Thank you**
> > > >
> > > > Thank you for the new experimental results, I believe this comparison strengthens your argument considerably, and will change my score to reflect it.

---

### Official Review · Reviewer_p6R1 · 2021-07-16

**Rating:** 6
**Confidence:** 3

**Summary:**

The paper presents an efficient gradient-based meta-learning algorithm based on evolutionary techniques. The proposed algorithm doesn’t require second order gradient computation, and thus, is more efficient. The proposed approach was evaluated on two meta-learning tasks for few-shot feature-wise transformation and noisy label learning, on which time and memory efficiencies were demonstrated.

**Limitations And Societal Impact:**

- The estimates for the meta-gradients could be noisy and suffer from high variance, it’s not clear how to solve this issue without increasing the population size which comes at an increased energy consumption, thus, the proposed approach presents a tradeoff between computing higher order derivatives, and possibly increasing the required memory and energy requirement.
 - As the authors discuss in the discussion section, the proposed approach is greedy and global convergence could not be guaranteed.
- The paper lacks theoretical analysis to the proposed approach.
- The empirical results are mostly limited to the computer vision domain, it’s not clear if the approach scales to larger models from the language and speech learning domains.

**Main Review:**

 # Originality

- The idea of using evolutionary methods to eliminate the need for higher-order gradients is novel.
- The related work is adequately cited, the proposed algorithm differs from the prior work in being more efficient in runtime and computation.

# Quality

- The submission is technically sound and the claims are supported by empirical results, but could be limited (please see below).
- The approach lacks the theoretical analysis for its effectiveness.
- The empirical analysis is limited to mostly computer vision applications.
- The proposed approach is appropriate for the studied applications, however, it’s not clear if the variance in the estimates for the meta-gradient would significantly affect the results for other applications.
- The authors are open about the limitations for their approach and discuss these challenges in the discussion section.

# Clarity

- The paper is clear, well organized, and easy to follow.

# Significance

- The work is significant as it aims at scaling meta-learning algorithms via using an evolutionary procedure to estimate meta-gradients without the need for computing higher order gradients.
- However, this estimate could be noisy and suffer from high variance (see Figure 2), it’s not clear how to mitigate this issue without increasing the population size leading to strong hits for the energy requirements for the proposed approach.

**Time Spent Reviewing:**

3

---

> ### Author Response · Authors · 2021-08-10
> **Response to the preliminary review**
>
> We appreciate the encouraging review and providing useful feedback. We respond to the comments below.
>
> **Q1. Theoretical analysis for effectiveness:** We provide a detailed theoretical analysis of how our method avoids higher-order gradients and hence is more efficient. Theoretical analysis of convergence rate is challenging in this case due to the stochastic nature of the approximation, and we leave this to future work. We argue this should not be a barrier to acceptance, as major competitors such as [18, 19] did not have convergence rate analysis at the time of publication.
>
> **Q2: Focus on computer vision (CV) applications:** We have selected applications which are popular (based on citations) and have code available. In fact, meta-learning has been applied more intensively in CV than other areas, so this has led to our specific focus. For other domains such as language and speech, we have found that meta-learning applications with publicly available implementations typically focus on few-shot learning (FSL). We do not study FSL because FSL already enjoys existing first-order approximations, and no alternative strategy is needed. We will continue searching for a suitable non-CV application and give a further update if we succeed within the discussion period. Nevertheless, we believe the current results already clearly show the benefits of EvoGrad, even if limited to CV, which on its own is already a large field and hence a significant impact for EvoGrad.
>
> **Q3. Variance in the estimates and other applications:** We believe our method is a valuable tool for the community even if it cannot be guaranteed before-hand to work well in every application considered. Many important tools in the deep learning community share this situation (e.g. dropout or label smoothing). Note that the noise can be tuned by modifying the values of temperature and epsilon - we provide simple guidance for how to select these in the appendix.
>
> **Q4. Greediness:** Please note that the greedy property of EvoGrad is exactly the same as the major competitor T1-T2, which we compare throughout. This is a common property of many practically applicable algorithms for meta-learning.

---

> > ### Comment · Reviewer_p6R1 · 2021-08-31
> > **Thanks for the clarifications and additional experiments**
> >
> > I thank the authors for their clarifications and the additional experiments. I still think the paper is above the acceptance threshold and I’m keeping my original score.

---

> ### Author Response · Authors · 2021-08-28
> **New experiments: EvoGrad on a NLP task**
>
> Thank you very much for suggesting to evaluate EvoGrad on a task that is outside of computer vision. We have added a new comment that describes evaluation of EvoGrad on a task from natural language processing. The results are positive and we are grateful you have suggested it as a way to better evaluate EvoGrad and show its significance.

---

### Official Review · Reviewer_xeM1 · 2021-07-16

**Rating:** 8
**Confidence:** 3

**Summary:**

This paper presents EvoGrad, a technique that replaces the inner loop of meta learning with an evolutionary technique to reduce the cost of meta learning - allowing meta learning to be applied to larger models. The paper demonstrates the EvoGrad can allow scaling up LFT from ResNet18 to ResNet34 and reduces the computation cost on MWN models.

**Limitations And Societal Impact:**

As described in the paper - the accuracy of the hyper gradient approximation could depend on the domain EvoGrad is applied to -  in the paper, all experimental tasks were related to image based data.

**Main Review:**

I vote to accept the paper.

The paper is clear and concise, the idea is neat and novel, the details are presented clearly and the experiments evaluate the efficacy of the method.

Originality: The idea of doing away with second-order gradients using a simple evolutionary technique in the inner loop is novel and, to the best of my knowledge hasn't been tried before. Second-order gradients are indeed responsible for the bottle neck in computation costs and getting rid of this computation entirely while retaining the accuracy is a pretty neat result.

Quality: The main contribution of EvoGrad is retention of performance and reduction in computation costs. These claims are substantiated as shown in Figures 4 and 5 and Tables 4 and 5. Moreover, the authors were able to scale up LFT to ResNet34.

As the authors mention, the hyper gradient estimate could be much worse on a different suite of problems. An excellent version of this paper would have evaluated the technique on more meta learning tasks. Alternatively, the paper could evaluate EvodGrad on task that is unrelated to computer vision to demonstrate that the hyper gradient approximation generalizes across domains.

Significance: If the hyper gradient approximation generalizes well to other meta learning tasks, the community will benefit from EvoGrad in that they will be able to use meta learning on larger models - an important contribution.

A minor comment on presentation - Figure 3 is not very readable - at first glance it wasn't clear to me if the three pictures are even different.

Typo in line 70
> or even a a synthetic training

**Time Spent Reviewing:**

2

---

> ### Author Response · Authors · 2021-08-10
> **Response to the preliminary review**
>
> Thank you for your encouraging review and in particular the comments about the significance of the improvements brought by the method. We have focused on the computer vision domain because existing online meta-learning methods focus predominantly on this domain as is evident in [9] (note that we also need publicly available PyTorch implementation of the approaches, which restricts to what problems we can compare). EvoGrad is a general method that is applicable to various meta-learning scenarios, but even if the approach only worked within computer vision it would have a huge impact on the community.
>
> Thanks for your editorial comments. We will address these.

---

> ### Author Response · Authors · 2021-08-28
> **New experiments: EvoGrad on a NLP task**
>
> Thank you very much for suggesting applying EvoGrad on a task that is unrelated to computer vision. We have added a new comment that describes evaluation of EvoGrad on a task from natural language processing. The results are positive and we are grateful you have suggested it as a way to make our paper excellent and further highlight the significance of EvoGrad.

---

> > ### Comment · Reviewer_xeM1 · 2021-08-28
> > **Thank you**
> >
> > Thank you for performing these additional experiments, I have adjusted my score.

---

### Official Review · Reviewer_7RkH · 2021-07-20

**Rating:** 6
**Confidence:** 4

**Summary:**

The authors introduce EvoGrad, a method for approximating second order gradients for use in Meta-Learning applications. The method approximates the inner loop of optimization with "evolutionary" update. The authors show their method is more computationally efficient than the T1-T2 method, yields good approximations to the second order gradients and is able to converge in the toy setting, and achieves better results than MetaWeightNet, a second order method, on a cross-domain few-shot classification task.

**Limitations And Societal Impact:**

Yes, there is a good paragraph addressing limitations and societal impact.

**Main Review:**

The main idea of the paper is not that technically novel but the paper is well written and easy to understand  -- explanations are clear and straight-forward.. The evolutionary update consists of a weighted sum of randomly selected perturbations of the current parameter values and reminds me of the finite differences approximation, although not an approximation to the first order gradient in this case. Instead, the first order gradient is removed from the update procedure so that optimizing hyperparameters does not require second order gradients. The weights for the parameter sum are selected based on a softmax procedure using their individual losses giving more weight to better performing parameter sets.

I found the computational arguments consisting of time requirements, memory requirements and scalability convincing.

I think I am misunderstanding something about the plot in figure 2: the red curve represents the ground truth hypergradient but the curve does not match the closed form hypergradient function defined on line 192.

The experiments analyzed in sections 4.1 and 4.2 use bounded hypergradients. Figure 2 seems to indicate that as lambda gets smaller the error between the approximation and the true hypergradient gets larger but, for this example, it's not too much of an issue since lambda is bounded below. Could we see divergence if the approximation error gets too large in unbounded cases?

I found the experiment section was good but there are still a few questions left unanswered in my opinion:
- The inner-loop is replaced only for the hyperparameter gradient. How good is the model that is fully trained using the "evolutionary update", ie. actually uses that inner-loop during training?

- How well does this model perform to the "first-order" approximations used by MAML and many other MAML-based models where higher order gradients are ignored via a stop-gradient procedure in the automatic differentiation software? Depending on which hyperparameter you are optimizing, this procedure would have different interpretations but would be easy to implement in all cases.

- How well does this compare to IFT methods? The authors mentioned that IFT methods assume that the inner-loop is run until convergence and that "this makes them unsuitable for the majority of practical applications above where training the inner loop to convergence for each hypergradient step is infeasible." However, without more explanation I don't see why this would be the case especially since EvoGrad uses an inner-loop approximation as well. Also, the authors state that in IFT methods, the "costs come from the associated overhead with approximation an inverse Hessian" but that this Hessian doesn't have to be stored in memory. However, what is required is the computation of Hessian-vector products which is a much cheaper computation.

-  In section 4.4, why does EvoGrad have better performance in terms of accuracy compared to MetaWeightNet? Isn't EvoGrad designed to be an approximation to the second order method used in MetaWeightNet? Is there something about the inductive bias in the approximation that would yield better results?

Due to these questions, I think this paper is marginally above the threshold of acceptance but I look forward to reading the opinions of the other reviewers and the response from the authors.

**Time Spent Reviewing:**

2.5

---

> ### Author Response · Authors · 2021-08-10
> **Response to the preliminary review**
>
> We appreciate the encouraging review and useful feedback, and we are also grateful for the questions that allow us to have a deeper discussion about our method. We provide answers to the questions below.
>
> **Q1. Plot in Figure 2 (curve and closed form expression for hypergradient):** The hypergradient ground-truth curve shown does indeed match the closed form expression in L192 ((x − 1)/(x + 1)^3). To see this clearly, plot this function in the region for x from 0 to 2, e.g., on Wolfram Alpha. We selected the given region to visualize because the true function is well-behaved there, allowing us to showcase the quality of approximation given by EvoGrad.
>
> **Q2. Hypergradient approximation divergence in unbounded cases:** The results in Fig 2 suggest EvoGrad may underestimate the hypergradient value if its magnitude is large. The gap increases with increasing magnitude of the ground-truth hypergradient value. However, this is not a problem in practice and may even be beneficial. In most cases of healthy deep learning the gradients have relatively small values. If we have large gradients, one often needs to use e.g. gradient clipping or normalization to make the learning stable. So in practice, EvoGrad should not suffer and may even benefit from underestimating large hypergradient magnitudes. Experimental results show EvoGrad matches the performance of standard T1-T2 meta-learning (applied to MetaWeightNet and CD-FSL) using the same number of iterations as T1-T2. So in practice either there is no under-estimation, or it does not lead to a slow down.
>
> **Q3. Fully training with the "evolutionary update":** The performance would certainly be extremely poor in this case - at least for applications where evolution is clearly worse than backpropagation for conventional model training, such as vision and NLP. Our insight is that the inner and outer optimization need not be the same, which leads to the key innovation of a hybrid approach using evolution to enable efficient computation of hypergradients, while actually updating the base model efficiently and accurately with gradient descent and backpropagation.
>
> **Q4. Comparison to the first-order approximation:** Please note that a key contribution of our work is to provide an efficient solution for cases where the first-order approximation is zero and hence not useful. The first-order approximation of FO-MAML/Reptile/etc. is typically only available in the special case where the meta-parameter to optimize is the same as the model’s weights. In the general case where the meta-parameter is anything else (e.g., a loss reweighting network in MWN, or noise generator in LFT - see [9] for many other examples), then the first-order approximation is usually not available (L97-101). EvoGrad provides an efficient solution in these more general cases.
>
> **Q5. Comparison to IFT methods:** (1) To elaborate, we meant that the assumption underpinning the derivation of IFT-based methods is that the inner-loop training has converged and has zero gradient. However in most practical applications this is too costly to perform and the inner loop is not run until convergence, thus IFT’s assumptions are usually violated in practice. EvoGrad derivation does not require making this same assumption. (2) In practice the IFT algorithm is unstable, especially where this assumption is violated. We spent effort working with an IFT-based competitor, but failed to achieve meaningful results. (3) T1-T2 (the stable method that is typically used in practice for meta-learning) can be considered as a special case of IFT that approximates the inverse hessian by identity (see Table 6 in the appendix or Table 1 in [18]). Even though Hessian-vector products are much cheaper than storing Hessian in memory, avoiding any Hessian-related calculations by identity is even more efficient, and we compare EvoGrad with this T1-T2 variant that is most commonly used in practice. (4) We also note that there are cases where EvoGrad can apply and IFT is inapplicable. For example, when using margin loss, such as in triplet ranking, the training loss can easily be zero, in which case the Hessian is an all zero matrix, and IFT will provide no hypergradient.
>
> **Q6. Better performance of EvoGrad on MWN:** In several cases the confidence intervals between EvoGrad and original MWN overlap, so the difference may not be significant. More fundamentally, we emphasize that EvoGrad is *not* merely trying to approximate the second order T1-T2 method. EvoGrad and second-order T1-T2 methods each provide distinct approximations to the true hypergradient. The empirical results show that in some cases the EvoGrad approximation is better than the T1-T2 approximation.
>
> Thanks a lot again for the encouraging review and the questions - we believe we have addressed the questions as requested and thus hope that the reviewer can raise their score.

---

> > ### Comment · Reviewer_7RkH · 2021-09-01
> > **Re: Response to the preliminary review**
> >
> > Thanks to the authors for the comments.
> >
> > Q1. Thanks for clearing this up. Due to the scale, it didn't look like the function passed through zero at x=1 as required.
> >
> > Q4. Again, thanks for the clarification.
> >
> > Q5. "EvoGrad derivation does not require making this same assumption". It is true that your method does not require this particular assumption but your method makes another approximation: that the gradient is well approximated by the evolutionary computation (random perturbations). A priori, I'm not sure why one approximation is better than the other.
> >
> > Q6. I understand that EvoGrad is not approximating T1-T2 which itself is an approximation to the hypergradient. But in your MWN you use the true hypergradients for training, or am I mistaken? This is why I'm confused about the better performance of EvoGrad.

---

> > > ### Author Response · Authors · 2021-09-01
> > > **Thank you and follow-up response**
> > >
> > > Thank you for the further comments, we respond to them below.
> > >
> > > Q5: We agree that each method makes some assumptions, especially the following:
> > > * IFT: Converged inner loop (definitely violated in practice in real applications, which use it in a short-horizon way), and acceptable accuracy of the Neumann Series/CG approximation to the inverse Hessian.
> > > * EvoGrad: Short horizon, and accuracy of the evolutionary approximation to inner loop during hypergradient calculation.
> > > * T1-T2: Short horizon, identity approximation to the Hessian.
> > >
> > > It could be a topic of a further paper to develop the theoretical insights required to decide when to prefer any of these based on first principles. Empirically EvoGrad is equally effective and more efficient than T1-T2, which is the method that is most widely used in practical applications due to its previously greater efficiency and practical stability compared to IFT (empirically IFT often leads to poor learning in practice, perhaps due to the violated assumption of inner-loop convergence).
> > >
> > > Q6: In our MWN (and all other experiments) we only use the approximation of hypergradient given by EvoGrad and not the true hypergradient for training - the true hypergradient is mostly a theoretical concept and its value needs to be approximated in practical applications. It should not necessarily be surprising that EvoGrad could outperform T1-T2, since different approximations may have different empirical performance.
> > >
> > > To elaborate, the source of inexactness comes from different places in T1-T2 and EvoGrad. In T1-T2, the outer loop is inexact due to using an approximation of the Hessian, while in EvoGrad the inner loop is inexact due to using evolutionary update.
> > >
> > > |        | Inner loop       |               Outer loop                   |
> > > | :-----:| :---------------:| :------------------------------------------:|
> > > | T1-T2  | (Exact) gradient | (Inexact) gradient (Hessian approximation)  |
> > > |EvoGrad | (Inexact) ES     | (Exact) gradient                            |

---

### Author Response · Authors · 2021-08-28
**New experiments: EvoGrad on a NLP task**

Evaluating EvoGrad on a real-world task outside of computer vision was suggested by two reviewers to make our paper excellent. We found a recent meta-learning approach from NLP that uses gradient-based meta-learning based on T1-T2: *MetaXL: Meta Representation Transformation for Low-resource Cross-lingual Learning* [A]. We have extended MetaXL with EvoGrad and obtained positive results, highlighting the wide applicability of EvoGrad.

We have taken the official code provided by [A] and tried to replicate their experiments as closely as possible. We selected the named entity recognition (NER) task with English source language (WikiAnn dataset), which is one of the key experiments in the paper. The only change we made is a smaller batch size: 12 instead of 16 to fit into the memory of the largest GPUs that we have currently available.

All details are described in [A]. For EvoGrad we have selected the same hyperparameters as for the other tasks in our paper (two model candidates, $\sigma = 0.001$ and $\tau = 0.05$).

We provide the comparison in the following three tables. Table 1 shows that EvoGrad matches and in fact surpasses the average test F1 score of MetaXL with the original T1-T2 meta-learning method. Tables 2 and 3 show that EvoGrad significantly improves both memory and time consumption compared to MetaXL T1-T2. In all tables JT stands for joint training and represents a simple baseline that does not use any meta-learning or any of the additional components used by MetaXL.


Test F1 score in % for named entity recognition task. English source language. The first two rows are taken from [A], while our own runs are in the next three rows. EvoGrad clearly matches and even surpasses the performance of T1-T2.


| Method                   | qu    | cdo   | ilo   | xmf   | mhr   | mi    | tk    | gn    | Average   |
| :----------------------- | :---: | :---: | :---: | :---: | :---: | :---: | :---: | :---: | :-------: |
| JT (paper)               | 66.10 | 55.83 | 80.77 | 69.32 | 71.11 | 82.29 | 61.61 | 65.44 | 69.06     |
| MetaXL T1-T2 (paper)     | 68.67 | 55.97 | 77.57 | 73.73 | 68.16 | 88.56 | 66.99 | 69.37 | **71.13** |
| JT (our run)             | 59.75 | 49.19 | 79.43 | 68.85 | 68.42 | 89.94 | 61.90 | 69.44 | 68.37     |
| MetaXL T1-T2 (our run)   | 65.29 | 56.33 | 76.50 | 67.24 | 71.17 | 89.41 | 66.67 | 64.11 | 69.59     |
| MetaXL EvoGrad (our run) | 71.00 | 57.02 | 85.99 | 70.40 | 65.45 | 88.12 | 66.97 | 70.91 | **71.98** |


Maximum allocated memory in GB - in all cases obtained by our own runs of the methods. EvoGrad significantly improves over T1-T2.

| Method         | qu   | cdo  | ilo  | xmf  | mhr  | mi   | tk   | gn   | Average |
| :------------- | :--: | :--: | :--: | :--: | :--: | :--: | :--: | :--: | :-----: |
| JT             |  7.2 |  9.1 |  7.3 |  7.5 |  7.3 |  7.7 |  8.1 |  9.3 |  8.0    |
| MetaXL T1-T2   | 16.0 | 18.8 | 16.2 | 16.6 | 16.2 | 16.8 | 17.3 | 19.0 | 17.1    |
| MetaXL EvoGrad | 12.1 | 13.2 | 12.2 | 12.3 | 12.2 | 12.4 | 12.6 | 13.2 | 12.5    |

Total wall-clock time in seconds of the different experiments. EvoGrad significantly improves over T1-T2.

| Method         | qu     | cdo    | ilo    | xmf    | mhr    | mi     | tk     | gn     | Average |
| :------------- | :----: | :----: | :----: | :----: | :----: | :----: | :----: | :----: | :-----: |
| JT             | 1493.9 | 1497.4 | 1455.8 | 1396.9 | 1396.5 | 1312.7 | 1266.1 | 1482.0 | 1412.7  |
| MetaXL T1-T2   | 5538.2 | 5320.5 | 3705.3 | 5272.6 | 5183.7 | 4908.6 | 3992.8 | 4043.8 | 4745.7  |
| MetaXL EvoGrad | 3464.7 | 3490.1 | 3642.7 | 3385.8 | 3080.5 | 3537.6 | 3302.3 | 3679.2 | 3447.9  |

We believe these additional experiments provide clear evidence for the wide applicability of EvoGrad. We are grateful that the reviewers have suggested this extension as it highlights the significant impact EvoGrad can make.

[A] Xia et al., NAACL 2021, MetaXL: Meta Representation Transformation for Low-resource Cross-lingual Learning

---

### Decision · Program_Chairs · 2021-09-27

**Decision:**

Accept (Poster)

**Comment:**

All the reviewers agree on acceptance. This is a clear decision. The authors should take into account the reviewers' feedback to improve the paper for the camera-ready submission.